# Crop Proteomics under Abiotic Stress: From Data to Insights

**DOI:** 10.3390/plants11212877

**Published:** 2022-10-27

**Authors:** Rehana Kausar, Xin Wang, Setsuko Komatsu

**Affiliations:** 1Department of Botany, University of Azad Jammu and Kashmir, Muzaffarabad 13100, Pakistan; 2College of Agronomy and Biotechnology, China Agricultural University, Beijing 100193, China; 3Faculty of Environment and Information Sciences, Fukui University of Technology, Fukui 910-8505, Japan

**Keywords:** proteomics, crop, abiotic stress

## Abstract

Food security is a major challenge in the present world due to erratic weather and climatic changes. Environmental stress negatively affects plant growth and development which leads to reduced crop yields. Technological advancements have caused remarkable improvements in crop-breeding programs. Proteins have an indispensable role in developing stress resilience and tolerance in crops. Genomic and biotechnological advancements have made the process of crop improvement more accurate and targeted. Proteomic studies provide the information required for such targeted approaches. The crosstalk among cellular components is being analyzed by subcellular proteomics. Additionally, the functional diversity of proteins is being unraveled by post-translational modifications during abiotic stress. The exploration of precise cellular responses and the networking among different cellular organelles help in the prediction of signaling pathways and protein–protein interactions. High-throughput mass-spectrometry-based protein studies are now possible due to incremental advancements in mass-spectrometry techniques, sample protocols, and bioinformatic tools as well as the increasing availability of plant genome sequence information for multiple species. In this review, the key role of proteomic analysis in identifying the abiotic-stress-responsive mechanisms in various crops was summarized. The development and availability of advanced computational tools were discussed in detail. The highly variable protein responses among different crops have provided a wide avenue for molecular-marker-assisted genetic buildup studies to develop smart, high-yielding, and stress-tolerant varieties to cope with food-security challenges.

## 1. Introduction

Abiotic stresses include drought, salt, temperature, waterlogging, and nutritional deficiency/excess, all of which hamper plant growth and seed yield to a great extent. During waterlogging, all or almost all of the plants are covered with water and face unfavorable conditions such as low light intensity, restricted gas diffusion, and the effusion of soil nutrients [1]. Plant-growth-promoting rhizobacteria encode 1-aminocyclopropane-1-carboxylate (ACC) deaminase which cleaves the substrate of ACC to produce ketobutyrate and ammonia to mitigate the adverse effects of waterlogging [2]. According to forecasts, half of all crop loss is attributable to abiotic stresses. However, abiotic stresses cause a myriad of changes in the physiological, molecular, and biochemical processes operating in plants [3]. These stressors greatly limit the distribution of plants, alter growth/development patterns, and reduce seed yield [4]. Studying drought and its potential impacts on food security is crucial in a global context [5]. Drought induces oxidative stress through the overproduction of reactive-oxygen species (ROS), ultimately causing the cell membrane to rupture and stimulating various stress-signaling pathways, including ROS, mitogen-activated protein kinase, Ca^2+^, and hormone-mediated signaling. The key responses against drought stress are root development, stomatal closure, photosynthesis, hormone production, and ROS scavenging [6]. Sustainable genetic improvements could be achieved through functional genomic approaches and gene-modification techniques, such as the CRISPR/Cas9 system that aids the characterization of genes sorted out from stress-related candidate single-nucleotide polymorphisms (SNPs), quantitative trait loci (QTL), and potential candidates [6]. Similarly, microbial transgenes were used to improve the tolerance of crops towards abiotic stresses, to increase desirable traits, and to improve yield/food quality [7]. Recently, an interactive dynamic regulatory events miner (iDREM) was developed, which integrates several high-throughput multiomic layers, including proteomics, transcriptomics, miRNA-omics, and epigenomics [8]. This type of advanced data integration could lead to a regulated network and could identify significant master regulators.

Great advancements in integrative data analysis have opened doors for enriching biological information for all plant species. In this context, proteomics has emerged as an especially useful tool to decipher crop performance in a globally changing environment [9]. The total protein complement is highly dynamic in time and space in any cell, tissue, or organ. There is a plethora of protein modifications, interactions, and network constellations at play at any specific developmental stage or physiological condition in plants [10]. An unimaginable expansion of proteomics was observed, with continuous innovations and optimizations in techniques, protocols, equipment, and associated bioinformatic tools [11]. The last decade has yielded significant developments in the field of proteomics, especially in mass spectrometry (MS) and data analysis tools. In particular, a shift from gel-based to MS-based proteomics has been observed, providing a platform with which to discover protein atlases for all lifeforms [12]. Based on this fact, the study of MS-based proteomics has gained tremendous popularity and data.

The complexity of the proteins in plant cells, which exhibit thousands of proteins whose abundance varies by several orders of magnitude, makes it impossible to cover most of the crop proteins when using MS-based proteomics [13]. Protein functions are tightly regulated by their subcellular localization and dynamic alteration. Proteomics does not rely on conventional organelle purification and can provide detailed insights into the microenvironments of diverse subcellular compartments [14]. Subcellular proteomics has the potential to elucidate localized cellular responses and to investigate communications among subcellular compartments during plant development and in response to abiotic stresses [15]. Continued advances in subcellular proteomics are expected to greatly contribute to the understanding of the responses and interactions that occur within and among subcellular compartments during development and under stressful environmental conditions [16]. It is important to completely comprehend the underlying mechanisms at the molecular level to gain a holistic picture of the stress-related responses generated in crops.

Post-translational modifications (PTMs) are at the heart of many cellular-signaling events. PTM crosstalk usually serves as a fine-tuning mechanism to adjust cellular responses to the slightest changes in the environment [17]. PTMs can regulate protein activity and localization as well as protein–protein interactions in numerous cellular processes, leading to the elaborate regulation of plant responses to various external stimuli [18]. Emerging evidence from plants indicates that PTMs intimately regulate numerous developmental programs and responses to the environment [19]. PTMs, including dephosphorylation, phosphorylation, glycosylation, and ubiquitination, play roles in regulating abscisic acid (ABA) signaling, which facilitates crop adaptation to stressful environments [20,21]. The comprehension of PTMs in plants can lead towards the creation of stress-tolerant varieties in the future.

In this review, the proteomic techniques used for the analysis of abiotic stress tolerance in crops and the importance of PTMs/subcellular localization were summarized. The potential applications of advanced signaling pathways in plant acclimation against abiotic stress were discussed to cope with food-security challenges.

## 2. Proteomic Technology Adopted by Plant Sciences

A two-dimensional gel electrophoresis (2-DE) technique was employed using isoelectric focusing and discontinuous SDS-polyacrylamide gel electrophoresis (PAGE) [22]. Gel-based techniques included differential in-gel electrophoresis (DIGE) for high resolution, sensitivity, and reproducibility by eliminating gel-to-gel variation. Gel-free technologies included multidimensional protein identification technology (MudPIT) for peptide separation, isotope coding, isobaric tagging for relative and absolute quantification, stable isotope labelling of amino acids in a cell culture, isotope-coded protein labelling for peptide quantification, and label-free methods [23]. The proteomic workflow started from tissue/cell fractionation, protein extraction, purification, separation in 2-DE, in-gel digestion, quantification, MS analysis, and data evaluation using a computer through algorithms for protein identification/quantification, bioinformatic data analysis, databases, and repositories [11]. Peanuts were analyzed under nonreducing conditions (protein-centric) and MS-based proteomics, and resulted in 60 new proteins [24]. Technological advancements in proteomics have greatly improved the detection of missed proteins over time (Table 1).

The comparison of gel-based and MS-based analyses was performed using the same protein extracts from *Quercus ilex* cotyledons at different development stages. The results obtained showed that both platforms were complementary, showing the common and specific proteins that were identified in each case and leading to similar biological conclusions. However, a greater number of proteins was obtained with the shotgun approach, with a greater representation of metabolic pathways, which were not commonly observed in gel-based analyses such as gibberellin biosynthesis [25]. During the proteomic workflow, in-gel digestion of target-protein spots was required to proceed for MS analysis. The modified method, termed high-throughput in-gel digestion (HiT-Gel), was based on a 96-well plate format which resulted in a drastic reduction in labor intensity and sample handling. The direct comparison revealed that HiT-Gel reduced technical variation and significantly decreased sample contamination more than the conventional in-gel digestion method [29].

Most plant-based food allergens are proteins with biological functions involved in storage, structure, and plant defense. Targeted proteomics, such as selected/multiple reaction monitoring (SRM/MRM), was found very useful, especially in the case of gluten from wheat/rye/barley and allergens from lentil/soybean/fruit. The detection and quantification of allergenic proteins using MS-based proteomics are promising and contribute to greater accuracy [26]. Advanced gel-based proteomic techniques combined with in-gel digestion coupled with MS/MS (GeLC-MS/MS) [30] are fast and informative regarding useful phenolic-compound-associated proteins and food allergens.

A shotgun proteomic method using nanoLC-MS/MS for proteins in the plasma membrane and plasma membrane microdomain fractions was developed, and the obtained results were easily applicable to label-free protein semi-quantification [31]. Sugarcane stem proteins were comprehensively studied using the shotgun proteomic approach and nuclei-enrichment sample preparation, which resulted in a large number of protein groups identified under drought, including many low-abundant nuclear proteins as well [32]. The correlation analysis of RNA-Seq and quantitative isobaric tags for relative and absolute quantitation (iTRAQ) LC-MS/MS was performed in healthy and tobacco mosaic virus infected leaves of Brassica. Finally, a network of proteins associated with infection was deciphered, which consisted of heat-shock proteins, calcium-signaling pathways, transcription factors, and lipid-transfer proteins [27]. The identification of greater and maximum numbers of proteins using the latest techniques was needed for low copy number and small proteins, which were usually missed in previous techniques. The difference of proteins in crop responses to abiotic stresses could uncover protein regulatory mechanisms and explain how specific proteins function in stress tolerance. The weakness of this approach was the low accuracy and narrow identification range of traditional shotgun proteomics, which hindered the discovery of genuine regulators in functional plant studies.

Recent technological advances provide an alternative label-free proteomic approach: sequential window acquisition of all theoretical mass spectra (SWATH-MS). It is characterized by a data-independent acquisition (DIA) method followed by a novel targeted-data-extraction approach. This powerful tool is now gaining attention in plant sciences as well [28]. In order to obtain deep and high-quality coverage of proteins, proper extraction methods for various tissues from different plant species are required. A sodium deoxycholate method was developed as a cheap, effective, and highly reproducible approach for total protein extraction followed by SWATH-MS detection. Differential proteomic analysis by SWATH-MS was performed to identify proteins in two contrasting rice varieties against leaffolder herbivory resistance. A few important defensive enzymes, which were phenylalanine ammonia lyase, chalcone synthase, and lipoxygenase, were found to be accumulated in the resistant rice line only [33]. In maize seedlings, substantial differences in post-transcriptional and translational changes between salt-treated embryo and endosperm were noticed using SWATH-MS, which indicated that the elevation of ABA level was an endosperm-dependent process [34]. In peanuts, differentially abundant proteins were identified using LC-MS coupled with tandem mass tag (TMT), and it was found that homologous proteins enriched different biological processes between the two subgenomes A and B [35]. SWATH-MS represents a valuable tool for quantitative proteomics based on its ability to generate reproducible data with the added benefit of allowing reinterrogation of data to improve analytical performance.

Finally, protein–protein interactions are the basis of various cellular processes. For the identification of protein–protein interactions in plant cells, TurboID-based proximity labeling is a recently developed method that addresses many limitations of previous methods. It allows rapid proximity labeling of proteins in just 10 min at room temperature [36]. This has significant advantages over traditional approaches, such as the ability to capture weak or transient interactions.

## 3. Crop Subcellular Proteomics

Plant cells are characterized by a high degree of compartmentalization as well as by having diverse proteins and metabolites. It is direly needed to resolve plant metabolism to a subcellular level to clarify the relationship between metabolites and proteins [37]. Subcellular proteomics has the potential to elucidate localized cellular responses and investigate communications among subcellular components in plant growth under abiotic stresses [38]. The crop proteins of the annotated location database cropPAL (http://crop-pal.org/ accessed on 10 September 2022) developed a range of subcellular location resources, including a species-specific voting consensus for 12 plant crop species [39]. In this review, the construction of useful bioinformatic tools for crop subcellular proteomics was discussed. Additionally, the current methodology used for subcellular proteomics and the molecular mechanisms involved were summarized (Table 2).

### 3.1. Construction of Useful Bioinformatic Tool and Database for Crop Subcellular Proteomics

Global agriculture demands better crop growth and larger crop yields. The subcellular localization of proteins is a key element in determining protein function, and proper protein functioning is critical for better harnessing energy for yield and plant defense and for sustainability [45]. The knowledge of protein subcellular localization is highly important for both basic and applied research. The artificial-intelligence-based predictor pLoc bal-mPlant was developed for identifying the subcellular localization of plant proteins. Its performance is excellent, particularly in dealing with multi-label systems, and it was trained by an extremely diverse dataset in which some subsets were more than ten-times larger than the others [46]. STRING (http://string-db.org accessed on 10 September 2022) is a database for protein–protein interactions. It collected and scored evidence from a number of sources and used it to integrate all known and predicted associations between proteins for a wide coverage [47]. Membrane proteins are estimated to constitute about a quarter of all proteins encoded in plant genomes. Using computational tools, ARAMEMNON (http://aramemnon.uni-koeln.de accessed on 10 September 2022) was designed to compile various computational predictions for plant membrane proteins [48]. Computational predictions provide an efficient way to infer the subcellular localization of a protein.

Plant-mSubP predicted 11 single localizations and 3 dual locations in plant cells with high accuracy as compared to the previously described tools. It used several hybrid features, such as autocorrelation and quasi-sequence-order descriptors, to represent the protein [49]. To accurately predict mitochondrial proteins, MU-LOC (http://mu-loc.org accessed on 10 September 2022) was developed. It collected a comprehensive dataset of subcellular plant localizations and extracted features, including amino acid composition, the protein-position weight matrix, and trained predictors, using a deep neural network and a support vector machine [50]. For plant vacuole protein prediction, VacPred (www.deepaklab.com/vacpred accessed on 10 September 2022) was developed using various models and achieved a higher accuracy than the previously developed methods [51]. The availability and usefulness of such bioinformatic tools are highly recommended for research based on their performance. Developing useful predictors and tools for subcellular localizations of plant proteins saved time by using a single interface.

### 3.2. Methodology of Subcellular Proteomics

Subcellular proteomics includes steps aimed at purifying organelles. For the preparation and enrichment of mitochondrial protein fractions, mitochondria in soybean root tips were isolated using the QProteome Mitochondrial Isolation kit (Qiagen, Hilden, Germany), and 134 mitochondrial proteins were identified [52]. A systematic and targeted analysis was carried out on the mitochondrial proteins from wheat root. Mitochondria were isolated by Percoll gradient centrifugation, and 140 mitochondrial proteins were identified [53]. The purification of mitochondrial fractions is easy and time saving when using the kit method as compared to the Percoll gradient.

As the plasma membrane is the first extracellular site to sense biotic or abiotic stresses, an understanding of the dynamics of plasma membrane proteins facilitates the development of new strategies for stress tolerance in crops. Proteins were extracted from highly enriched plasma membrane fractions of chickpea seedlings using aqueous two-phase partitioning, and 2732 nonredundant proteins were identified [54]. Plasma membrane protein fractions from rice root, etiolated leaf, green leaf, developing leaf sheath, and flower were analyzed using the two-polymer phase system, and 511 commonly accumulated proteins were identified [55]. For the isolation of plasma-membrane-enriched fractions, an aqueous two-polymer phase system is the best method, which is applicable to a wide variety of plants with high purity levels.

The integrity of subcellular proteomics is largely dependent on the purity of the isolated compartment from other contaminants. Nuclei were enriched from soybean root tips, and the purity of enriched nuclei fractions was analyzed using immunoblot and enzyme activity assays, identifying 365 proteins [56]. A soybean subcellular root protein study was performed using fractions of membranes and nuclei, resulting in 441 identified proteins. Plant-derived smoke solution increased soybean root growth through transcriptional promotion with RNA polymerase II expression and energy production with ATPase accumulation [43]. The use of the nuclei isolation/extraction kit (Sigma-Aldrich, St. Louis, MO, USA) is an easy and convenient methodology employed in many recent studies.

A refined protocol for isolating endoplasmic reticulum and assessing purity was developed, in which an isosmotic homogenization buffer consisting of HEPES, sucrose, and KCl was used for soybean root tips [57]. In wheat seedling leaves, endoplasmic reticulum protein purification was performed using HEPES and DTT. In total, 234 differentially accumulated proteins were identified including disulfide-isomerase-like proteins, heat-shock proteins, and ribosomal proteins [58]. Previously, different hydrodynamic techniques, such as differential and density gradient centrifugation, were applied to isolate endoplasmic reticulum proteins. In more recent studies, improved purity has been assessed via electron microscopy and/or Western blotting for endoplasmic reticulum protein fractions using calreticulin, BiP, and calnexin as the markers.

An optimized method for chloroplast isolation increased the yield of tomato chloroplasts eightfold, enabling the proteomic analysis of the chloroplast-stromal proteins. It is a high-yielding method to isolate leaf chloroplasts and stromal proteins for subcellular proteomic studies. The set of 254 proteins that coisolated with the chloroplast stroma provided information for developing a better understanding of the extensive and dynamic interactions of chloroplasts with other organelles during abiotic stress [40]. Some chloroplast proteins are known to serve as messengers to transmit retrograde signals from chloroplasts to the nuclei in response to environmental stresses. In tobacco, under drought, 1087 differentially accumulated chloroplast proteins were identified [59]. Chloroplast subcellular proteomics can reveal the possible function of proteins used as messengers in retrograde-signaling pathways, which is essential to further understand the molecular mechanisms underlying abiotic stress in crops. Based on the available data, it is clear that efficient fractionation and enrichment protocols are needed for high-throughput subcellular-proteomic analysis.

To investigate the proteins of peroxisomes in plants, the free-flow-electrophoresis (FFE) technique was successfully used. Generally, the proteins on organelles sum up to a negative surface charge, which enforces their migration towards the anode in an electric field. Different organelles exhibit distinct membrane proteins, generating organelle-specific net surface charges. These properties were effectively exploited by the technique of FFE, whose instrument consisted of a separation chamber with a laminar buffer flow perpendicular to an electric field. Thus, individual particles were deflected from the linear flow in the separation chamber according to their electrophoretic mobility and isoelectric point [60]. Hence, the separation of similar-density particles was sorted out by using FFE for plant peroxisomes.

### 3.3. Subcellular Proteomics in Understanding Mechanisms in Crops under Abiotic Stress

Chloroplasts are critical subcellular organelles since they act as metabolic factories and the seat of photosynthesis that perceive metabolic and stress signals and convey them to different cellular components. They are key regulators of stress perception and signal transduction, and they are the site of the production of the secondary metabolites and plant hormones involved in defense during abiotic stress. An optimized method for chloroplast isolation increased the yield of tomato chloroplasts eightfold, enabling the proteomic analysis of the chloroplast–stromal proteins. It is a high-yielding method to isolate leaf chloroplasts and stromal proteins for subcellular proteomic studies. The set of 254 proteins that coisolated with the chloroplast stroma provided information for developing a better understanding of the extensive and dynamic interactions of chloroplasts with other organelles during abiotic stress [40]. Using soybean leaves, chloroplast analysis was performed using the proteomic technique. Various bioinformatic analyses were performed, and, in total, 3148 acetylation sites in 1538 chloroplast proteins were detected. Ribosome activity, protein biosynthesis, and fatty acid metabolism related proteins were the most preferred substrates in lysine acetylation in leaves. The acetylated proteins were mainly localized in the chloroplast and played an important role in soybean physiology and biology, referring to its functional and structural characterization in oil crops [59]. The crucial role of photosynthetic machinery during abiotic stress is highly dependent on the chloroplast proteins.

In maize, the superior vigor of hybrid seedling roots compared to their parental inbred lines was studied. Using the primary roots of maize, the subcellular fractions of cell walls, plasma membranes, and secreted mucilage were analyzed, resulting in the maize–root system being extensively developed to provide seedling vigor during seed development [44]. In another study of maize proteins, various *cis*-acting elements revealed the responsiveness of cell-wall protein-coding genes towards phytohormones and abiotic stress. Using bioinformatic tools, 20 stress-responsive proteins were found to be promising candidates for applications in developing stress-resistant maize varieties [42]. Resolving subcellular proteomics can lead to a better understanding of the molecular architecture of any specific organ or tissue at any developmental stage in crops under abiotic stresses.

## 4. Crop Proteomics of Post-Translational Modifications

PTMs are the dynamic and reversible modifications of proteins and have wide effects in the functionality of proteins. Specific proteins found within cells play crucial roles in stress mitigation by enhancing the cellular processes that facilitate plant survival during abiotic stresses. However, proteins that already exist in the plant cells can be subjected to an array of PTMs that permit an instant and rapid response. These activated proteins can, in turn, further aid in stress responses. Different PTMs perform different functions under abiotic stresses [61]. It is estimated that approximately 50% of all proteins are glycoproteins, of which the majority are *N*-glycosylated. The presence or absence of *N*- and *O*-glycans on glycoproteins was shown to influence the activity, stability, and functionality of proteins to a large extent, and played a critical role in cellular signaling, molecular trafficking, plant development, and adaptation to abiotic stresses [62]. Likewise, ubiquitination was involved in the regulation of a variety of processes in plants including the endocytic trafficking and vacuolar turnover of plasma membrane proteins. Additionally, phosphorylation and lysine acetylation were critical coregulators of nitrogen fixation in legumes [63]. These studies provide supportive evidence for the key role of PTMs in abiotic stress acclimation and tolerance (Table 3).

### 4.1. Importance of Post-Translational Modifications in Crops

Crops are food producers, and the size of plant organs is mainly determined by the coordination between cell expansion and proliferation [77]. The protease DA1 (One LIM Domain of Arabidopsis) limits the duration of cell proliferation and thereby restricts final organ size. The interacting proteins of DA1 include cleavage substrates and proteins, which modulate its activity through PTMs, such as ubiquitination, deubiquitination, and phosphorylation [78]. Furthermore, seed germination and seedling establishment are important developmental processes that undergo a plethora of physiological changes and are precisely regulated at transcriptional and translational levels. Phosphorylation and ubiquitination are involved in regulating the function of ABAI5 (ABA insensitive 5) transcription factor. The PTMs on ABAI5 provided a model that could be used to understand the stability and activity of the specific proteins modified during seed germination [79]. The involvement of such PTMs, such as phosphorylation, ubiquitination, and deubiquitination, during seed germination implies the potential crosstalk among various PTMs during this essential phase of seedling establishment.

Plant peptide hormones/growth factors consisting of 70–120 amino acids emerged as an important class of cell-to-cell signals for short-distance communication as well as for long-range signaling. Multiple PTMs were required for this peptide–hormone activation, and many of the enzymes responsible for these modifications were identified [80]. PTMs allow cells to transduce signals, regulate protein functions, and respond to cellular disturbances/perturbations. They expand protein functionality and diversity, which leads to increased protein complexity. PTM crosstalk explains the connective action of multiple PTMs on the same or on different proteins for a higher order regulation in plant cells [81]. The PTM crosstalk focuses individual protein molecules as signaling hubs which are involved in the integration of post-translational signals from multiple sources.

### 4.2. Post-Translational Modifications in Understanding Mechanisms in Crops under Abiotic Stress

Protein glycosylation is involved in physiological functions and biological pathways such as protein folding, glycan-dependent quality control processes in the endoplasmic reticulum, protein stability, and protein–protein interactions. To understand the involvement of *N*-glycoproteins in the mechanism of the drought response in leaves of the common bean, a proteomic approach using lectin affinity chromatography, SDS-PAGE, and LC-MS/MS was applied. Label-free quantification of *N*-glycoproteins was performed using MaxQuant. Beta-glucosidase showed the highest increase in abundance among proteins involved in cell-wall metabolism, indicating the positive role of glycosylation in cell-wall modification under drought stress [64]. The *N*-glycoproteomic analysis of the plasma membrane in wheat under drought stress was performed using HILIC enrichment and LC-MS/MS. *N*-glycosylation sites were concentrated within the [N × T] motif, and 79.5% of them were located on the random coil, which facilitated protein glycosylation and enhanced structural stability during drought stress [65]. The role of protein glycosylation in strengthening the cell wall and remodeling the plasma membrane is evident during abiotic stresses such as drought.

Protein phosphorylation is commonly involved in key regulatory processes which mediate plant growth and development during abiotic stress responses. The molecular mechanism of the crop response to chilling stress was uncovered through the study of protein phosphorylation. In rice, CNGC9 (cyclic nucleotide-gated channel 9) is activated by a dehydration-responsive element-binding transcription factor, DREB1A. The loss-of-function of *Oscngc9* suggested that CNGC9 enhanced the chilling tolerance in rice by regulating cold-induced calcium influx and cytoplasmic calcium elevation [66]. Phosphoproteomics is the tool of choice for analyzing phosphorylation-signaling networks because it can reveal these modifications in an unbiased manner. System-wide phosphoproteomic analyses led to the identification of over 1000 phosphoproteins in various plant species, providing crucial insights into the regulation of protein functions [82]. The phosphoproteomic changes of two tomato cultivars with distinct cold-tolerance phenotypes were profiled under cold stress. Unique phosphopeptides from tomato leaves were identified, and the data was confirmed after validation through in vitro kinase reactions. The confirmed phosphokinases were involved in cold-tolerant signaling [67]. Maize leaves are a common model used to study the C4 photosynthesis mechanism. Identified phosphoproteins indicated the role of the phosphorylation of cell-wall metabolism related enzymes in the regulation of the transition from proliferative cell division to cellular differentiation. During de-etiolation of maize seedlings, most phosphoproteins were found to be involved in gene transcription, post-transcriptional regulation, and signal transduction [68].

Protein ubiquitination is involved in various biological processes including proteolysis and the endocytic trafficking and vacuolar turnover of plasma-membrane proteins. In maize kernels, the specific K-GG antibody coupled with high-resolution LC-MS/MS was used to identify the function of ubiquitinated proteins. Eight conserved ubiquitinated motifs were found in ubiquitinated peptides, and proteins involved in carbohydrate metabolism, protein processing, RNA transport, spliceosomes, endocytosis, ubiquitin-mediated proteolysis, proteasomes, and MAPK signaling were the main targets [69]. In another study, the functional analysis of the tomato carboxyl terminus of *CHIP (HSC70-interacting proteins)* ubiquitin E3 ligase was performed under heat stress. The accumulated protein aggregates were highly ubiquitinated, and the results indicated that the tomato CHIP played a critical role in the heat-stress response most likely by targeting the degradation of misfolded proteins that were generated during heat stress [70]. As a chaperone-dependent E3 ubiquitin ligase, CHIP targets the ubiquitination of misfolded proteins for degradation through the 26S proteasome system.

Cysteine *S*-nitrosylation is a reversible PTM that critically regulates the activity, localization, and stability of proteins. A global analysis of cysteine *S*-nitrosylation in tea leaves was performed, and the *S*-nitrosylated proteins were found to be located in various subcellular compartments, especially in the chloroplast and cytoplasm. Furthermore, the analysis of functional enrichment and the protein–protein interaction network revealed that *S*-nitrosylated proteins were mainly involved in multiple metabolic pathways, including glycolysis, pyruvate metabolism, the Calvin cycle, and the tricarboxylic acid (TCA) cycle [72]. Among 866 acetylated proteins of rice seedlings, 38 acetylation sites were combined in core histones. Protein–protein interaction networks of the identified proteins provided further evidence, showing that the acetylation level of histone H3 (lysine 27 and 36) was increased in response to cold stress [76]. In soybeans, the proteomic analysis of *S*-nitrosylated proteins was performed under flooding stress. The pathway mapping of the *S*-nitrosylation protein profile indicated a characteristic pattern in glycolysis and fermentation, which was confirmed by a Western blot analysis. *S*-nitrosylation increased the active form of alcohol dehydrogenase under flooding [71]. It is imperative to note that PTMs act as the background triggering factors responsible for all metabolic shifts and adaptations in plants.

Protein malonylation is one of the PTMs that plays a critical role in the diverse metabolic processes in both eukaryotes and prokaryotes. In wheat, the malonylated proteins were located in multiple subcellular compartments, mostly in the cytosol and chloroplast. The protein–protein interaction network analysis revealed highly interconnected clusters of malonylated proteins and out of which 137 proteins were mapped in the protein network database. Moreover, few proteins were simultaneously modified by lysine malonylation, acetylation, and succinylation, suggesting that these three kinds of PTMs may coordinately regulate the function of the proteins involved in the carbon fixation in wheat [74]. A qualitative analysis to globally identify malonylated proteins in maize was also performed in a study. The great number of uniquely malonylated lysine residues was observed in different proteins. A large proportion of modified proteins was located in the chloroplast and was involved in photosynthesis and the Calvin cycle, suggesting an indispensable regulatory role of malonylation in photosynthesis and carbon fixation [83]. It is suggested with evidence that the role of malonylation is quite important in further deciphering the Calvin cycle in plants.

The lysine 2-hydroxyisobutyrylome analysis was performed in wheat roots with antibody immunoprecipitation affinity, a high resolution of MS-based proteomics, and bioinformatics. The functional and metabolic characterization of the modified lysine 2-hydroxyisobutyrylation proteins revealed that various cellular functions and metabolism pathways were potentially affected by them [84]. In another study, lysine 2-hydroxyisobutyrylation sites of modified proteins were identified in rice seedlings. A motif analysis revealed conserved motif-flanking sites localized in the chloroplast. Gene ontology, the KEGG pathway, and protein domain enrichment analyses confirmed that lysine 2-hydroxyisobutyrylation was on proteins involved in diverse biological processes and was especially enriched in carbon metabolism and photosynthesis [73]. For regulating protein functions, lysine 2-hydroxyisobutyrylation allowed the modified lysine to form hydrogen bonds with other biomolecules that were involved in key metabolic pathways.

## 5. Crop Proteomics in Understanding Environmental Stress Responses

Abiotic stresses reveal profound impacts on plant proteins, including alterations in relative protein abundance, cellular localization, PTMs, and protein–protein interactions [85]. In plants, multiple responses are induced in response to environmental stress and allow rapid adjustment of the abundance and function of key stress-response components. Crop responses to abiotic stresses are dynamic and intricate as well as varied with the type, level, and term of stress [86]. The global vulnerability of crops towards abiotic stresses has tremendously increased due to climate change. Using drought-tolerant rice cultivar, label-free MS-based proteomics was performed. Leaves from the drought and well-watered controls were harvested. The biochemical and physiological analyses confirmed that the drought-tolerant response in rice leaves was unique with superior leaf water status and enhanced levels of photosynthesis-related proteins [87]. A comparative proteomic approach was applied to analyze the protein changes of embryo and endosperm during seed germination in winter wheat salt-tolerant cultivar. The identified proteins were found to be involved in stress defense, energy, and protein/amino acid metabolism. Proteins related to storage and starch metabolism were more abundant in endosperm, while the embryo had unique proteins involved in lipid and sterol metabolism [88]. Flooding-stress tolerance in soybeans was enhanced in the mutant line compared to the parental type. The acquisition of flood tolerance was due to decreased cell death along with glycoprotein folding in the mutant line [89]. The information about crop-tolerance mechanisms against abiotic stresses is the key to unraveling the potential target protein/gene cascade patterns involved in stress acclimation and tolerance (Appendix A, Figure 1 and Figure 2).

### 5.1. Rice Proteomics in Understanding Signaling Mechanisms under Drought Stress

Rice is a critical nutritional food source, but its yield is vulnerable to periods of drought. Different genotypes of upland and lowland rice were exposed to drought stress at the late vegetative stage, and leaves were harvested for label-free MS-based proteomics. Heat-shock- and late-embryogenesis-associated proteins were found to be induced in response to drought stress in all genotypes. The drought resilience mechanism in diverse rice genotypes is an indication of the broad genetic base available for further exploitation to drought-tolerant breed varieties [87]. In a study of rice spikes under drought stress, proteins were identified using iTRAQ LC-MS/MS. The inhibition of the photosynthetic rate was mainly through stomatal conductance restriction and flavonoid pathway regulation of ROS during stress, which ultimately led to the decreased yield. Energy and ROS metabolism were also badly affected under the condition of drought [90]. CRISPR/Cas9 generated rice OsPYL9, which are pyrabactin-resistance-like protein mutants, showed an increase in grain yield under both drought and well-watered field conditions due to the accumulated proteins related to circadian clock rhythm, ROS, drought responsive ABA signaling, and protein–protein interactions [91]. The availability and utilization of the genetic diversity and genomic techniques have now opened up an era of exploring new avenues for drought stress resilience in crops.

The exogenous spray of CPPU (N-2-(chloro-4-pyridyl)-N-phenyl urea), which is a synthetic cytokinin, improved the stomatal conductance in leaves of drought-sensitive rice cultivar. Among the photosynthetic pigments, chlorophyll *b* contents were significantly reduced by drought, whereas CPPU-treated plants retained the normal contents of chlorophyll *b* under stress. These results indicated a drought-mediated suppression of chlorophyll synthase and 7-hydroxymethyl chlorophyll a reductase [92]. Total DNA methylation increased in rice plants when drought was applied at the vegetative stage but decreased in plants stressed at the reproductive stage. The first drought event induced adaptation to water-deficit conditions through decreasing energy dissipation, increasing ATP energy provision, and reducing oxidative damage in photosynthesis and energy metabolism. The stress memory was also associated with epigenetic markers [93]. The drought tolerance in rice could be enhanced by overexpressing *PeaT1*, which is the protein elicitor. The abundance of 18 proteins was detected in proteomic data such as OsSKIPa and OsPP2C, which were significantly induced in the early stage after dehydration treatment in transgenic rice [94]. The application of various growth enhancers and the exploration of genes related to drought tolerance proved to be an efficient strategy for increasing drought tolerance in rice.

### 5.2. Wheat Proteomics in Understanding Signaling Mechanisms under Salt Stress

Wheat is the worldwide staple crop for food security, and salinity is a major abiotic stress that negatively affects its growth and development. MS-based proteomic analysis was applied to decipher the proteomic profiling of the germinating wheat seeds subjected to salt treatment. Significant alteration in protein abundance, related to phenylpropanoid biosynthesis, fatty acid degradation, oxidative phosphorylation, RNA degradation, glyoxalate, and dicarboxylate metabolism, was observed. At the transcriptional level, lignin biosynthesis varied greatly during salt stress in germinating wheat seeds [95]. MS-based proteomic analysis identified that proteins involved in light-dependent reactions decreased under salt stress along with a significant upregulation of proteins associated with the Calvin cycle, plastoglobule development, protein folding/proteolysis, and hormone/vitamin synthesis. These molecular changes indicated that salt stress was more deleterious to wheat seedlings compared to osmotic stress [96]. Salt stress downregulated the chloroplastic proteins involved in light-dependent reactions in wheat seedlings.

The proteomic technique was employed to identify the differentially abundant proteins in wheat roots in response to salt treatment using an absolute quantitation-based method. The observed salt-responsive proteins were ubiquitination-related proteins, transcription factors, pathogen-related proteins, membrane-intrinsic protein transporters, and antioxidant enzymes [97]. In another study, wheat plants were treated with salt stress and then proteomic analysis was carried out on the control and stressed plants. It was observed that metabolism- and photosynthesis-related proteins increased, while proteins related to the TCA cycle were decreased [98]. The accumulation of malate in wheat roots was found to be a crucial adaptive mechanism during salt stress. The observed decline in energy and photosynthesis-related proteins during salt stress is a clear indication of its role in the reduced yield of wheat.

### 5.3. Maize Proteomics in Understanding Signaling Mechanisms under Osmotic Stress

Drought strictly impacts the growth, development, and yield of maize crops. Although it is evident through transcriptomic analysis that the transcriptional factors, such as BRI1-EMS suppressor 1/brassinazole-resistant 1, negatively regulated drought tolerance in maize [99], this hormonal regulation was not established by proteomic analysis. The proteomic analysis of water-deficit maize leaves identified proteins were mainly involved in photosynthesis, carbohydrates, stress defense, energy production, and protein metabolism [100]. In the drought-tolerant maize variety, the accumulated proteins in the leaves repaired photosystem II and improved photochemical capacity. The response of maize shoots under drought stress involved proteins related to photosynthesis, amino acids, sugar/starch metabolism, and redox regulation [101]. The phosphoproteomic analysis in maize revealed that serine/threonine protein phosphatase 2C (ZmPP2C26) altered the phosphorylation level of proteins involved in photosynthesis [102]. Furthermore, leaf proteins involved in redox homeostasis and the electron transport chain helped in drought tolerance [103]. The drought-tolerance mechanism in maize roots was attributed to proteins involved in stronger water-retention capacity, activation of antioxidant enzymes, strengthening of the cell wall, osmotic stabilization of the plasma membrane, effective recycling of amino acids, and improved lignification [104]. Leaf proteins involved in improved photosynthesis and root proteins related to osmotic stabilization, lignified cell wall, and amino acid recycling confer drought tolerance in maize.

Salt stress in maize triggers complex physiological processes, resulting in osmotic disturbance and stunted growth [105]. Maize transcriptomic studies revealed that brassinosteroid-signaling transcription factors (ZmBES1/BZR1-5) positively regulated salt tolerance [106]. The adaptation of maize seedlings to salt stress involved ROS scavenging, nitrogen/glutamate metabolism, energy homeostasis, nucleotide transport, soluble sugars, fatty acids, and nucleoside triphosphates synthesis [107]. In maize leaves, the differentially regulated proteins functioned in growth-relevant processes and in cell-wall building components during salt stress [108]. Meanwhile, in maize roots, the positively correlated proteins for stress tolerance were involved in oxidation reduction, catabolic processes, chemical stimulus responses, and translational elongation [109]. Differentially regulated proteins, such as glucose-6-phosphate 1-dehydrogenase, NADPH-producing dehydrogenase, glutamate synthase, and glutamine synthetase, were accumulated in salt-tolerant maize-seedling roots [110]. Maize tolerance to salt stress involves cell-wall building proteins in leaves, and it involves pentose-phosphate/glutathione metabolism and nitrogen metabolism in roots.

### 5.4. Soybean Proteomics in Understanding Signaling Mechanisms under Flooding Stress

Soybean is an important oil seed crop whose growth is severely impaired by flooding stress. The promoting effects of melatonin on soybeans under flooding stress were studied using proteomic analysis. RNA regulation and cell-wall/protein metabolism were enriched by identified proteins and transcripts encoding proteins that respond to flooding stress and melatonin treatment. Activated cell degradation, expanded intercellular spaces, and reduced lignification in the root tips of flooded soybeans were ameliorated by melatonin treatment [111]. Protein quality control and calcium homeostasis were found to be disrupted in the endoplasmic reticulum of soybeans under flooding stress. Decreased ribosomal proteins were responsible for suppressed protein synthesis, but the heterogeneity in ribosomal proteins displayed a different selectivity [112]. The soybean flooding response mechanism identified proteins were essentially involved in cell-wall-, ribosomal-, mitochondrial-, and protein-degradation metabolism.

The mechanism underlying the tolerance of flooding stress in soybeans was also explored through MS-based proteomic studies. A soybean mutant line was subjected to flooding for 2 days, and proteins were analyzed. Immunoblot analysis confirmed that calnexin accumulation increased in both the wild type and mutant line, while calreticulin accumulated in only the mutant line under stress. The flooding tolerance in the mutant line was enhanced due to cell death regulation through the fermentation system and glycoprotein folding in the endoplasmic reticulum [89]. Millimeter-wave irradiation improved the growth of the root and hypocotyl of soybeans during flooding stress. Proteomic analysis indicated the recovery of irradiated seedlings under stress, whereas proteins related to glycolysis and ascorbate/glutathione metabolism were not affected. Sugar metabolism was suppressed under flooding, whereas it was improved in the irradiated soybean plants, especially in trehalose synthesis [113]. In another study of soybeans under flooding, the root and hypocotyl length/weight of soybeans were enhanced by silver nanoparticles mixed with nicotinic acid and by KNO_3_ application. The accumulation of calnexin/calreticulin and glycoproteins increased, and caused positive effects on soybean seedlings by regulating the protein quality control for the misfolded proteins in the endoplasmic reticulum [114]. Plant-derived smoke enhanced soybean growth during recovery from flooding stress through the balance of glycolysis and sucrose/starch metabolism. The protein abundance and gene expression of *O*-fucosyltransferase family proteins related to the cell wall were higher in smoke-treated flooded soybeans than in plant-flooded soybeans alone. The protein abundance and gene expression of peptidylprolyl cis-trans isomerase and the Bowman–Birk proteinase inhibitor D-II were lower in smoke-treated flooded soybeans than in flooded soybeans [41]. The interplay of various proteins and metabolites during the flooding tolerance mechanism in soybeans needs to be understood completely for progression towards the creation of flooding-tolerant soybean lines.

## 6. Future Perspectives

Proteomics is an advanced and high-throughput technique, which provides in-depth information about intricate molecular mechanisms in plants. Crops have developed various adaptations and strategies to cope with the changing climatic conditions. Highly advanced instrumentation and separation techniques have made it possible to fully understand complicated protein combinations and their interactions within and outside the cell. Advanced gel-free/label-free proteomics, newly developed genetic/genomic procedures, and an array of highly advanced bioinformatic tools have now made it possible to thoroughly understand complex protein/gene operational cascades in different crops. The recent studies of subcellular proteomics and PTMs have greatly helped with comprehending the complex linkage between stress tolerance and enhancing crop productivity. The discovery of potential protein biomarkers in crops is another strategy, which could be used for efficient crop-breeding improvement programs. The utilization of gene-editing techniques, such as CRISPR/Cas9 technology, is also very useful in targeting potential stress-related genes. The development of stress-tolerant and high-yielding crop varieties is direly needed to ensure food security (Figure 3). Taken together, data from genomics, transcriptomics, proteomics, and metabolomics can better lead us towards a comprehensive understanding of the multiple factors playing roles in changing climate. All these studies will be able to jointly catalyze/enhance marker-assisted selection in crops for the development of stress-tolerant and high-yielding varieties under abiotic stresses.

## Figures and Tables

**Figure 1 plants-11-02877-f001:**
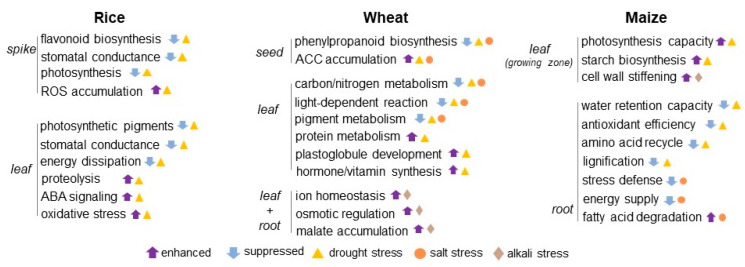
Crop proteomics in understanding signaling mechanisms under osmotic stress. The schematics of crop response to osmotic stress was constructed based on the proteomic studies on rice, wheat, and maize under drought, salt, and alkali conditions. The upward and downward arrows mean enhanced and suppressed metabolisms induced by osmotic stress. Triangle, circle, and diamond indicate drought, salt, and alkali stress, respectively. ABA stands for abscisic acid; ACC stands for acetyl-CoA carboxylase; and ROS stands for reactive oxygen species.

**Figure 2 plants-11-02877-f002:**
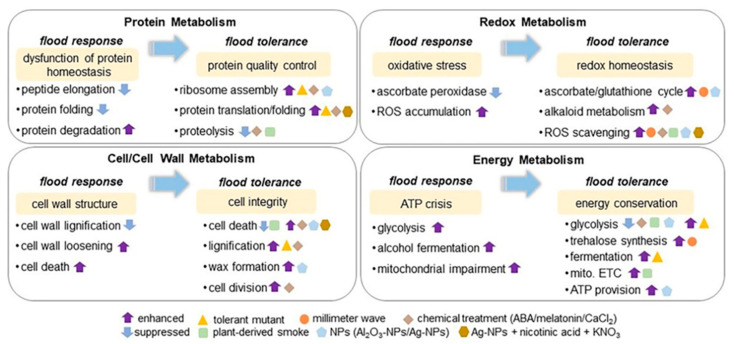
Soybean proteomics in understanding signaling mechanisms under flooding stress. The presented schemes of core-signaling mechanisms, including protein metabolism, redox metabolism, cell/cell-wall metabolism, and energy metabolism, involved in stress response and tolerance in the early-stage soybean under flooding stress was summarized based on proteomic studies. The signaling mechanism of flooding tolerance was constructed from proteomic data collected from soybean plants with flood-tolerant characteristics. The metabolisms highlighted with yellow color mean induced events for stress response and tolerance in flooded soybean. The upward and downward arrows mean enhanced and suppressed metabolisms. Triangle indicates flood-tolerance-mutant of soybean developed from gamma irradiation. Circle, diamond, square, pentagon, and hexagon indicate the treatment of millimeter-wave irradiation, chemical application (abscisic acid, melatonin, and CaCl_2_), plant-derived smoke, nanoparticles (NPs), and Ag-NPs with nicotinic acid and KNO_3_, respectively, alone with flooding stress. ABA stands for abscisic acid; mito. ETC stands for mitochondrial electron transport chain; and ROS stands for reactive oxygen species.

**Figure 3 plants-11-02877-f003:**
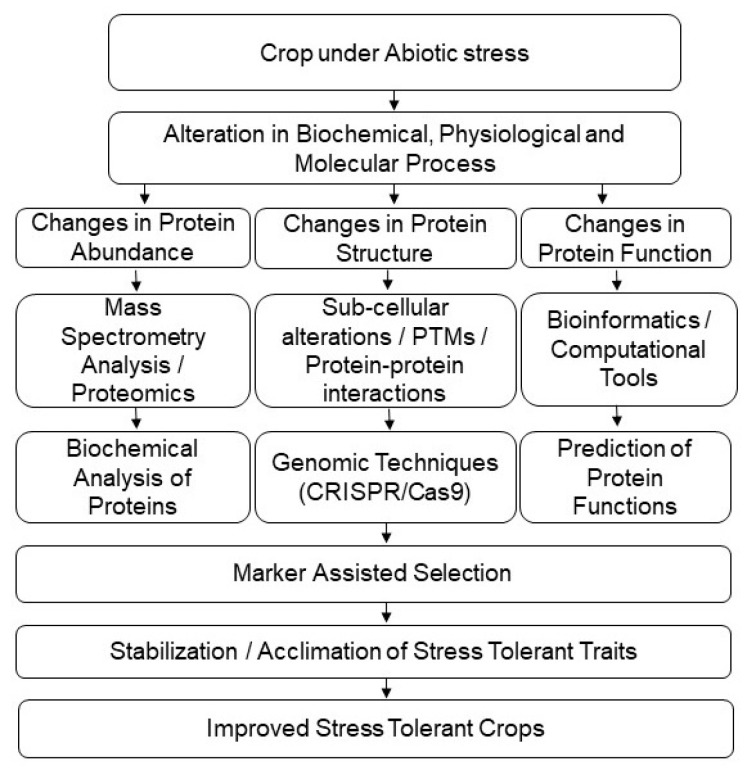
Application of proteomic techniques for crop improvement under abiotic-stress conditions.

**Table 1 plants-11-02877-t001:** Different techniques used in crop proteomics.

Technique (Labeling Tool)	Detection Tool	Merits/Demerits	Stage *	Ref **
DIGE (Cy3, Cy5)/MALDI-TOF/MS	TOF	Time and cost effective/Not suitable for hydrophobic proteins	First	[23]
MudPIT	Conversion dynode/electron multiplier	Less time consuming/PTMs cannot be detected	Second	[23]
LC-MS/MS	ESI	Quicker with less extensive extraction procedures and high resolution/More time consuming	Second	[11]
MALDI-TOF/MS	TOF	Fast, accurate, and less expensive/Low analytical sensitivity	Second	[25]
GeLC-MS/MS	ESI-MS/MS	High speed with excellent quantification properties/Less sensitive for large proteins	Third	[25]
LC-MS/MS	SRM/MRM	Valuable tool for biomarker validation/Different ionization of individual peptide modifications	Third	[26]
iTRAQ-LC-MS/MS	Multiplex stable isotope amino-specific reagent	High-throughput multiplexing capacity/Time consuming, laborious, and expensive	Third	[27]
SWATH-MS	DIA	Comprehensive quantitative analysis/Hard data analysis requiring sophisticated software tools	Fourth	[28]

* Stage refers to generation stage: first (gel-electrophoresis-based), second (based on isobaric or isotopic labelling), third (label-free, gel-free, or shotgun), and fourth (mass-western, targeted) generation techniques. ** Ref means reference. Abbreviations: 2-DE stands for two-dimensional electrophoresis; IEF stands for isoelectric focusing; DIGE stands for differential gel electrophoresis; MudPIT stands for multidimensional protein identification technique; MALDI-TOF stands for matrix-assisted laser desorption and ionization time of flight; LC-MS/MS stands for liquid chromatography–mass spectrometry/mass spectrometry; GeLC-Orbitrap/MS stands for gel-enhanced liquid chromatography–orbitrap mass spectrometry; SRM/MRM stands for selective reaction monitoring/multiple reaction monitoring; ESI stands for electrospray ionization; iTRAQ stands for isobaric tags for relative and absolute quantitation; and SWATH-MS stands for sequential window acquisition of all theoretical fragment ion spectra–mass spectrometry.

**Table 2 plants-11-02877-t002:** Subcellular proteomics employed in crop research during 2017–2022.

Subcellular	Crop	Organ	Purification Method	MS Methodology	No *	Major Findings	Ref **
Stroma, chloroplast	Tomato	Leaf	Percoll reagent	2D-LC-MS/MS	254	Optimized method for chloroplast isolation increased the number of tomato chloroplasts eightfold.	[40]
Chloroplast, ribosome	Soybean	Leaf	NETN buffer, anti-acetyl lysine antibody beads	EASY-nLC-MS/MS	1538	Motif analysis of modified peptides extracted 17 conserved motifs of acetylation.	[41]
Cell wall	Maize	Leaf/Root	Vacuum-infiltration–centrifugation technique	ESI-qQ-TOF-MS/MS	863	Twenty cell-wall proteins were declared as potential candidates against both biotic and abiotic stresses.	[42]
Plasma membrane, nuclei	Soybean	Root	Mem-PER plus extraction kit, plant nuclei extraction kit	LC-MS/MS	268	ATPase increased in plasma membrane, and nuclear proteins mainly decreased. However, RNA polymerase II was upregulated.	[43]
Cell wall, plasma membrane, secreted mucilage	Maize	Root	CHAPS with TCIP buffer	2D-LC-MS/MS	150	Eight lateral root initiation mutant-specific proteins were identified, out of which four were involved in lignin metabolism.	[44]

* No represents number of identified proteins. ** Ref stands for reference.

**Table 3 plants-11-02877-t003:** PTM proteomics in crops during 2017–2022.

PTMs	Crop	Organ	PTM Detection Method	MS	No *	Major Findings	Ref **
Glycosylation	Common bean	Leaf	Qproteome Total Glycoprotein Kit	LC-MS/MS	35	Beta-glucosidase increased among proteins which were involved in cell-wall metabolism.	[64]
	Wheat	Leaf	Hydrophilic interaction liquid chromatography (HILIC) enrichment	LC-MS/MS	173	Glycosylated proteins (related to protein kinase activity involved in the reception and transduction of extracellular signals and plant cell-wall remolding) were regulated by drought stress.	[65]
Phosphorylation	Rice	Protoplast	HIS-kinase buffer with an anti-FLAG antibody	LC-MS/MS	152	Cyclic-nucleotide-gated channel OsCNGC9 enhanced chilling tolerance in rice through regulating cold-induced Ca^2+^ influx and cytoplasmic Ca^2+^ elevation.	[66]
	Tomato	Leaf	PolyMAC-Ti kit	LC-MS/MS	550	The activation of SnRK2s and their direct substrates assisted tomatoes in surviving long-term cold stress.	[67]
	Maize	Leaf	Nickel–nitriloacetic acid beads	LC-MS/MS	692	Tyrosine phosphorylation and calcium signaling pathways played important roles during de-etiolation of leaves.	[68]
Ubiquitination	Maize	Kernel	PTM Scan ubiquitin remnant motif K-ε-GG kit	LC-MS/MS	881	Eight conserved ubiquitination motifs, including KubD, GKub, Ekub, KubXXXE, Akub, NXKub, KubXXXXXN, and Kkub, were found in ubiquitinated peptides.	[69]
	Tomato	Leaf	Anti-ubiquitin monoclonal antibody	LC-MS/MS	652	Tomato carboxyl terminus of hs70-interacting proteins played a critical role in heat stress response most likely by targeting degradation of misfolded proteins that were generated during heat stress.	[70]
*S*-nitrosylation	Soybean	Seedlings	*S*-nitrosylated-protein detection assay kit	LC-MS/MS	162	Western blot analysis confirmed that *S*-nitrosylated status of alcohol dehydrogenase increased with flooding.	[71]
	Tea	Leaf	Anti-TMT Resin	LC-MS/MS	191	RuBisCO was *S*-nitrosylated at 6 cysteine sites, and *S*-nitrosylated-ALDO played crucial role in Calvin cycle and glycolysis.	[72]
	Rice	Seedling	Antinitryl antibody beads	LC-MS/MS	866	Nitrosylated proteins were involved in histone H3, and different sites were combined in core histones to enhance chilling stress.	[73]
Lysine malonylation	Wheat	Leaf	Antimalonyllysine antibody agarose beads	LC-MS/MS	233	Protein interaction network analysis revealed eight highly interconnected clusters of malonylated proteins which were involved in carbon fixation.	[74]
Lysine acetylation	Soybean	Leaf	Agarose beads eluted by trifluoroacetic acid	LC-MS/MS	17	The conserved motifs of lysine-acetylated peptides were extracted, which were found to be involved in ribosome activity and protein biosynthesis.	[75]
	Rice	Leaf	Anti-acetyl lysine antibody beads and NETN buffer	LC-MS/MS	866	Eleven lysine motifs were conserved, and 45% of the identified proteins were localized in chloroplast; 38 lysine acetylated motifs were combined in core histones.	[76]

* No represents number of identified proteins. ** Ref stands for references.

## Data Availability

Not applicable.

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
