# Peer review of "Crop Proteomics under Abiotic Stress: From Data to Insights"

_plants, 2022, doi:10.3390/plants11212877_

Round 1

Reviewer 1 Report

The manuscript by Kausar and Komatsu is aiming at summarizing proteomic techniques used for the analysis of abiotic stress tolerance in crops and the importance of posttranslational modifications/subcellular location (for stress tolerance?). However, most of the text consists of relatively general statements or is summarizing findings from various studies, but without synthesizing and integrating the knowledge from these studies. Also, it touches upon many different topics, but without going into detail with any of them, and the text is lacking a red thread. The overall purpose of the review remains unclear to me, and it is difficult to see how the draft adds anything to several recent reviews on similar topics.

The introduction works relatively well, although it is not clear why there are some lines focusing on drought stress specially (lines 35-40).

Lines 93-95: Some introductory text is needed. In which study was 2-DE technique employed? It says gel electrophoresis is an excellent complementary and alternative approach, but as an alternative to what?

Lines 139-143: Not clear why the results from reference no 26 are summarized?

Lines 144-155: Summary of finding from various studies, however, the reason for summarizing these studies is not clear.

Lines 161-169: Results from two studies employing new proteomic approaches are summarized, however, it remains unclear what are the weaknesses and strengths of these approached compared to other approaches or how do the results differ from what would be expected using older approaches.

Lines 187-189: You can hardly call Table 2 a description of various subcellular proteomics covering the dimensions of subcellular proteomics. The table list some studies focused on subcellular proteomics, but there is no description and the suitability of various approaches is not clear.

Paragraph 3.1: The manuscript is describing four protein localization prediction tools, but many more are available. Why focus on the four mentioned?

Paragraph 3.2: This paragraph also contains a description of various methodologies used in different studies focusing on subcellular proteomics. However, the reasons for summarizing these methodologies are not clear (how do they differ, which methodologies are best for various purposed, what are the strengths and weaknesses of various approaches etc.)

My impression of the rest of the manuscript is similar to the impression reflected in the comments above; I don´t think the review to a sufficient extent describe the relationship of each work summarized to the others under consideration.

Author Response

Reviewer 1

The manuscript by Kausar and Komatsu is aiming at summarizing proteomic techniques used for the analysis of abiotic stress tolerance in crops and the importance of posttranslational modifications/subcellular location (for stress tolerance?). However, most of the text consists of relatively general statements or is summarizing findings from various studies, but without synthesizing and integrating the knowledge from these studies. Also, it touches upon many different topics, but without going into detail with any of them, and the text is lacking a red thread. The overall purpose of the review remains unclear to me, and it is difficult to see how the draft adds anything to several recent reviews on similar topics.

Answer: Thank you very much for your critical comments. This article has been corrected as suggested. These corrections have been improved with new Figures 1 and 2.

The introduction works relatively well, although it is not clear why there are some lines focusing on drought stress specially (lines 35-40).

Answer: Drought stress was included in the introduction, because studying drought and its potential impacts on food security is very crucial in a global context. Based on this reason, drought stress was explained as an example. Further edits have been made in the introduction section as suggested.

Lines 93-95: Some introductory text is needed. In which study was 2-DE technique employed? It says gel electrophoresis is an excellent complementary and alternative approach, but as an alternative to what?

Answer: Thank you very much for your comments. Upon further consideration, because there are many limitations of 2-DE for protein identification, the related reference and sentence have been removed.

Lines 139-143: Not clear why the results from reference no 26 are summarized?

Answer: We are sorry for the lack of clarity. This sentence has been corrected in red as follows: “Advanced gel based proteomic techniques combined with in-gel digestion coupled with MS/MS (GeLC-MS/MS) [26] are fast and informative regarding useful phenolic associated proteins and food allergens.” This is why the reference 26 was summarized.

Lines 144-155: Summary of finding from various studies, however, the reason for summarizing these studies is not clear.

Answer: These studies were summarized, because the identification of greater and maximum number of proteins using latest techniques is needed for low copy number and small proteins, which were usually missed in previous techniques. Further edits have been made in this section per your comments.

Lines 161-169: Results from two studies employing new proteomic approaches are summarized, however, it remains unclear what are the weaknesses and strengths of these approached compared to other approaches or how do the results differ from what would be expected using older approaches.

Answer: Thank you very much for your comments. As suggested, this paragraph has been corrected as follows: “The weakness of this approach was the low accuracy and narrow identification range of traditional shotgun proteomics, which hindered the discovery of genuine regulators in plant-functional studies.” “In order to obtain deep and high-quality coverage of proteins, proper extraction methods for various tissues from different plant species are required. A sodium deoxycholate method was developed as a cheap, effective, and highly reproducible approach for total protein extraction followed by SWATH-MS detection.”

Lines 187-189: You can hardly call Table 2 a description of various subcellular proteomics covering the dimensions of subcellular proteomics. The table list some studies focused on subcellular proteomics, but there is no description and the suitability of various approaches is not clear.

Answer: We are sorry for this problem. This summary of section 3 was not good explanation and this summary has been corrected.

Paragraph 3.1: The manuscript is describing four protein localization prediction tools, but many more are available. Why focus on the four mentioned?

Answer: We are sorry for this problem, again. Paragraph 3.1 has been rewritten with additional two current tools in red.

Paragraph 3.2: This paragraph also contains a description of various methodologies used in different studies focusing on subcellular proteomics. However, the reasons for summarizing these methodologies are not clear (how do they differ, which methodologies are best for various purposed, what are the strengths and weaknesses of various approaches etc.)

Answer: Thank you very much for your comment. Paragraph 3.2 has been corrected as suggested; which how do they differ, which methodologies are best for various purposes, and what are the strengths/weaknesses of various approaches.

My impression of the rest of the manuscript is similar to the impression reflected in the comments above; I don´t think the review to a sufficient extent describe the relationship of each work summarized to the others under consideration.

Answer: This article has been corrected in its entirety. We hope you find that this article to be more suitable for this journal now.

Reviewer 2 Report

The manuscript titled “Crop Proteomics under Abiotic Stress: From Data to Insights” reported proteomic techniques used for the analysis of abiotic stress tolerance in crops and the importance of PTMs/subcellular localization are summarized. This is a well-written article and I anticipate that the manuscript should be of great interest. Before recommending this article for publication, there are some shortcomings that should be resolved.

General comments

Overall, the study is well designed and presented in a good way. However, the authors are suggested to read the manuscript carefully for the correction of typo mistakes. 

Introduction

Line 30 replace “flooding” with “waterlogging” and Cite the following articles in line 31.
https://doi.org/10.3389/fmicb.2018.01096

https://doi.org/10.1007/s13765-018-0392-2

Furthermore, cite the following article in line 44.

https://doi.org/10.4014/jmb.1808.08029

Put a relevant reference to lines 39 and 40.

Delete “_” in line 61.

Keep the same format for subheadings.

Add a comprehensive figure which represents a schematic presentation of Crop proteomics and abiotic stress.

Author Response

Reviewer 2

The manuscript titled “Crop Proteomics under Abiotic Stress: From Data to Insights” reported proteomic techniques used for the analysis of abiotic stress tolerance in crops and the importance of PTMs/subcellular localization are summarized. This is a well-written article and I anticipate that the manuscript should be of great interest. Before recommending this article for publication, there are some shortcomings that should be resolved.

General comments

Overall, the study is well designed and presented in a good way. However, the authors are suggested to read the manuscript carefully for the correction of typo mistakes. 

Answer: We apologize for the typographical errors in the article. The typographical errors have been corrected in this article.

Introduction

Line 30 replace “flooding” with “waterlogging” and Cite the following articles in line 31.
https://doi.org/10.3389/fmicb.2018.01096

https://doi.org/10.1007/s13765-018-0392-2

Answer: Thank you very much for your critical suggestion. Sentences with the two suggested references have been added in the introduction section. The added references are “Ali et al.,2018a” and “Ali et al., 2018b”.

Furthermore, cite the following article in line 44.

https://doi.org/10.4014/jmb.1808.08029

Answer: Thank you very much for suggesting it. The reference “Ali et al., 2018c” has been cited in text.

Put a relevant reference to lines 39 and 40.

Answer: Reference has been included as “Mahmood et al., 2019”.

Delete “_” in line 61.

Answer: We are sorry for this mistake. It has been deleted.

Keep the same format for subheadings.

Answer: The format for subheadings has been checked and corrected in red.

Add a comprehensive figure which represents a schematic presentation of Crop proteomics and abiotic stress.

Answer: Thank you very much for your suggestion. Table 4, including its many references, has been shifted to Table S1 and new Figures 1 and 2 have been prepared.

Reviewer 3 Report

General comments:

The review titled Crop Proteomics under Abiotic Stress: From Data to Insights”, reviewed technological advancements that caused remarkable improvements in the crop-breeding process as well as summarized the key role of proteomic analysis in identifying the abiotic stress-responsive mechanisms in various crops to cope with food-security challenges. The review was well-written and easy to follow. However, I have some suggestions, as shown below:

1, The manuscript summarizes the recent update on crop proteomics under Abiotic Stress and further reviewed the recent method for plant proteomics, however, it seems to miss one advanced protein labeling method: TurboID-based proximity labeling. It is a recent update on protein proximity labeling in plants, highly suggest adding it to the review.

2, Hormone-related Abiotic Stress should be deep discussed and in part 5, the author reviewed the most important crops but missing maize here, it already has the studies one the Hormone-related Abiotic Stress in maize (PMID: 35682705, PMID: 32028614, PMID: 35463404).

3, Too many tables in the text and highly suggested schematic figures to clearly show the review.

 Minor revision:

Line 15: crosstalk

Line 31: what is, according to the projection means

Line 61: Additional “_” here

Line168: in peanut,

Line 327: ABI5 (abscisic acid insensitive 5)

Author Response

Reviewer 3

The review titled “Crop Proteomics under Abiotic Stress: From Data to Insights”, reviewed technological advancements that caused remarkable improvements in the crop-breeding process as well as summarized the key role of proteomic analysis in identifying the abiotic stress-responsive mechanisms in various crops to cope with food-security challenges. The review was well-written and easy to follow. However, I have some suggestions, as shown below:

1, The manuscript summarizes the recent update on crop proteomics under Abiotic Stress and further reviewed the recent method for plant proteomics, however, it seems to miss one advanced protein labeling method: TurboID-based proximity labeling. It is a recent update on protein proximity labeling in plants, highly suggest adding it to the review.

Answer: Thank you very much for your valuable suggestion. Explanation of TurboID-based proximity labeling has been added in the text. The reference is “Zhang et al., 2020”.

2, Hormone-related Abiotic Stress should be deep discussed and in part 5, the author reviewed the most important crops but missing maize here, it already has the studies one the Hormone-related Abiotic Stress in maize (PMID: 35682705, PMID: 32028614, PMID: 35463404).

Answer: Thank you very much for your kind suggestion. The new section “5.3 Maize Proteomics Understanding Signaling Mechanism under Osmotic Stress” has been added with suggested references as well as additional references. Furthermore, that information has been written in new Table S1.  

3, Too many tables in the text and highly suggested schematic figures to clearly show the review.

Answer: Thank you very much for your suggestion. Table 4, including its many references, has been shifted to Table S1 and new Figures 1 and 2 have been prepared.

Minor revision:

Line 15: crosstalk

Line 31: what is, according to the projection means

Line 61: Additional “_” here

Line168: in peanut,

Line 327: ABI5 (abscisic acid insensitive 5)

Answer: Thank you very much for your pointing these out. They have been corrected in the text in red. Furthermore, all parts in this text have carefully corrected.